# Mutual-Information Regularized Multi-Agent Policy Iteration

**Jiangxing Wang**
School of Computer Science
Peking University
jiangxiw@stu.pku.edu.cn

**Deheng Ye**
Tencent Inc.
dericye@tencent.com

**Zongqing Lu**[†]
School of Computer Science
Peking University
BAAI
zongqing.lu@pku.edu.cn

## Abstract

Despite the success of cooperative multi-agent reinforcement learning algorithms, most of them focus on a single team composition, which prevents them from being used in more realistic scenarios where dynamic team composition is possible. While some studies attempt to solve this problem via multi-task learning in a fixed set of team compositions, there is still a risk of overfitting to the training set, which may lead to catastrophic performance when facing dramatically varying team compositions during execution. To address this problem, we propose to use mutual information (MI) as an augmented reward to prevent individual policies from relying too much on team-related information and encourage agents to learn policies that are robust in different team compositions. Optimizing this MI-augmented objective in an off-policy manner can be intractable due to the existence of dynamic marginal distribution. To alleviate this problem, we first propose a multi-agent policy iteration algorithm with a fixed marginal distribution and prove its convergence and optimality. Then, we propose to employ the Blahut–Arimoto algorithm and an imaginary team composition distribution for optimization with approximate marginal distribution as the practical implementation. Empirically, our method demonstrates strong zero-shot generalization to dynamic team compositions in complex cooperative tasks.

## 1 Introduction

The cooperative multi-agent reinforcement learning (MARL) problem has attracted the attention of many researchers for being a well-abstracted model for many real-world problems, such as traffic signal control (Wang et al., 2021), autonomous warehouse (Zhou et al., 2021), and even AutoML (Wang et al., 2022) as the feedback to the machine learning community. In a cooperative MARL problem, we aim to train a group of agents that can cooperate to achieve a common goal. Such a common goal is often defined by a global reward function that is shared among all agents. Although this objective is naturally centralized, we want agents to be able to execute in a fully decentralized manner. Under such a requirement, Kraemer and Banerjee (2016) proposed the centralized training with decentralized execution (CTDE) framework, where a centralized critic is learned to evaluate the performance of the joint policy in terms of the global reward and a group of decentralized individual policies are learned via the centralized critic to realize decentralized execution.

With a centralized critic, multi-agent policy gradient methods directly use it to guide the update of each decentralized individual policy. Based on this idea, a series of studies (Lowe et al., 2017; Kuba et al., 2022; Yu et al., 2022; Ye et al., 2020) have been proposed with different optimization

---

[†]Corresponding Author

37th Conference on Neural Information Processing Systems (NeurIPS 2023).

techniques for policy improvement. On the other hand, as the centralized critic is used to guide the learning of decentralized individual policies, many CTDE algorithms choose to factorize the centralized critic into decentralized individual utilities via the mixer network. This line of research is called value decomposition (Sunehag et al., 2018). Based on different design choices of the mixer network, a variety of value decomposition methods (Sunehag et al., 2018; Rashid et al., 2018; Wang et al., 2020; Zhang et al., 2021) has been proposed and achieved great success in cooperative MARL problems.

Despite the success of CTDE methods, previous research mainly focuses on training agents under a fixed team composition, which has been shown to exhibit serious overfitting issues (Wen et al., 2022), and leads to catastrophic performance when facing unseen team compositions. One natural way to solve this problem is to introduce multiple team compositions during training to alleviate overfitting. However, due to the existence of the mixer network and the network structure of individual utilities and policies, such a strategy cannot be simply applied to many existing CTDE methods.

REFIL (Iqbal et al., 2021) attempts to address this problem and proposes to use multi-head attention (Vaswani et al., 2017) in the mixer network and individual utilities to handle dynamic team compositions. It further proposes an imaginary objective based on random sub-group partitioning to accelerate the training process given a fixed set of team compositions. Although REFIL is able to handle dynamic team compositions, as it is still trained on a fixed set of team compositions, there is still a risk of overfitting to not a single team composition, but a set of team compositions, which has been demonstrated in several previous studies (Liu et al., 2021; Shao et al., 2022).

The overfitting issue can be attributed to the lack of robustness in different team compositions. When facing an arbitrary team composition during execution, one agent's observed information about team composition can arbitrarily vary from what it experienced during training. If the agent puts too much credit on this highly varied information to make decisions, it may fail to achieve robust behavior. As the variation of team composition is uncontrollable, one way to achieve robust behavior is to reduce the reliance on team-related information. Based on this intuition, we propose **MIPI** (**M**utual-**I**nformation Regularized Multi-Agent **P**olicy **I**teration), minimizing the mutual information between the policy of the agent and the team-related information to encourage robust behavior of each agent. Inspired by SAC (Haarnoja et al., 2018), we combine the global reward of environment and mutual information of each agent as a new objective and learn individual policies to optimize them at the same time. As the incorporation of mutual information imposes a challenge on optimization due to the existence of dynamic marginal distribution, we first propose a multi-agent policy iteration algorithm with a fixed marginal distribution and prove its convergence and optimality. Then, we propose to utilize the Blahut–Arimoto algorithm (Cover, 1999) and an imaginary team composition distribution for optimization under an approximate dynamic marginal distribution as the practical implementation.

To empirically justify our algorithm, we first evaluate the performance of MIPI in a simple yet challenging matrix game. Compared with the other two baselines using pure environmental reward and entropy-augmented reward, using mutual information as an augmented reward can help the agent find the policy that can achieve consistent performance across different team compositions. Then, we move to a more complicated scenario, StarCraft Micromanagement Tasks. While having the same level of performance in the training set, MIPI achieves better zero-shot generalization results in unseen team compositions during evaluation.

## 2 Related Work

### 2.1 Centralized Training with Decentralized Execution (CTDE)

CTDE methods can be categorized into value decomposition and multi-agent policy gradient, depending on whether a centralized critic is decomposed or not. For value decomposition methods, a centralized critic is decomposed into decentralized utilities through the mixer network. Different mixers have been proposed as different interpretations of the Individual-Global-Maximum (Rashid et al., 2018) (IGM) principle or its equivalence, which ensures the consistency between optimal local actions and optimal joint action. VDN (Sunehag et al., 2018) and QMIX (Rashid et al., 2018) give sufficient conditions for IGM by additivity and monotonicity, respectively. QPLEX (Wang et al., 2020) and FOP (Zhang et al., 2021) take advantage of duplex dueling architecture to guarantee IGM.

In multi-agent policy gradient, a centralized critic function is directly used to guide the update of each decentralized individual policy. Most multi-agent policy gradient methods can be considered as an extension of the policy gradient method from RL to MARL. For example, MAPPDG (Lowe et al., 2017) extends DDPG (Lillicrap et al., 2016), HATRPO (Kuba et al., 2022) extends TRPO (Schulman et al., 2015), MAPPO (Yu et al., 2022) and CoPPO (Wu et al., 2021) extend PPO (Schulman et al., 2017).

## 2.2 Dynamic Team Composition

While classical CTDE methods mainly focus on fixed team compositions, in real-world applications, agents that can adapt to dynamic team composition are preferable. To address this problem, RE-FIL (Iqbal et al., 2021) incorporates multi-head attention (Vaswani et al., 2017) into the networks and further introduces an imaginary objective based on random sub-group partitioning to accelerate the training process on a fixed set of team compositions. While REFIL learns policies that can handle dynamic team compositions, studies (Liu et al., 2021; Shao et al., 2022) suggest that it generalizes poorly on unseen team compositions. The necessity of training agents that can generalize to unseen team compositions is evident using the automated warehouse as an example, where it is very common to add more agents (more agents got purchased) or delete some agents (some agents got broken). Therefore, agents have to deal with different teams in the application scenario, which can not be fully covered during training. In order to learn policies that can adapt to unseen team compositions, many studies choose to sacrifice the requirement of decentralized execution. For example, a centralized agent is assumed in COPA (Liu et al., 2021), which has a global view of the environment and coordinates agents by distributing individual strategies. In SOG (Shao et al., 2022), a communication channel is assumed to elect conductors, so that the corresponding groups are constructed with conductor-follower consensus. Unlike these methods, we do not assume any kind of centralization during execution, such that the ability of decentralized execution is fully preserved. In CollaQ (Zhang et al., 2020), the decentralized utility function is decomposed into two terms: the self-term that only relies on the agent's own state, and the interactive term that is related to states of nearby agents. By using an additional MARA loss to constrain the contribution of the interactive term in the decentralized utility function, CollaQ solves the generalization on the dynamic team composition problem with CTDE being preserved. Unlike this method, MIPI uses mutual information to constrain the contribution of team-related information in the agent's policy.

## 2.3 Information-Theoretic Principles in RL

In the standard RL problem, the objective is to solely optimize the environmental reward. However, in many problems, we not only want to optimize the cumulative rewards but also want the learned policy to exhibit some other properties. Therefore, a line of research across single-agent and multi-agent domains has been proposed using different information-theoretic principles for policy optimization. For example, SQL (Haarnoja et al., 2017) and SAC (Haarnoja et al., 2018) incorporate the maximum entropy principle to encourage exploration and diverse behavior. FOP (Zhang et al., 2021) further extends this idea to MARL and proves its convergence and optimality. As the augmented entropy term distorts the original objective in the original MDP, which may lead to undesirable behavior in some scenarios (Eysenbach and Levine, 2019). To solve this problem, DMAC (Su and Lu, 2022) proposes to use the divergence between the current policy and previous policy to replace entropy, which yields a bound of the discrepancy between the converged policy and optimal policy in the original MDP. Using divergence to guide the policy optimization is also very popular in offline RL (Levine et al., 2020), for example, F-BRC (Kostrikov et al., 2021) and ICQ (Yang et al., 2021), where the divergence is used to control the similarity between learned policy and behavior policy. While the entropy and KL divergence can be seen as a measurement of the distance to a fixed policy, the mutual information is about the distance to a dynamic marginal policy. In MIRL (Grau-Moya et al., 2019) and MIRACLE (Leibfried and Grau-Moya, 2020), the environmental reward is combined with mutual information to encourage the learned policy to be close to an optimal prior policy, which is also dynamically learned during the RL process instead of being fixed throughout. Unlike (Grau-Moya et al., 2019; Leibfried and Grau-Moya, 2020) that aim at a generalized version of SAC in single-agent RL, the purpose of our work is to solve generalization on dynamic team composition in MARL.

Mutual information (MI) has been widely used in previous MARL research for various purpose, e.g., exploration (Mahajan et al., 2019; Wang et al., 2019; Zheng et al., 2021), coordination (Konan et al., 2021; Kim et al., 2023), individuality (Jiang and Lu, 2021), diversity (Li et al., 2021) and social

influence (Jaques et al., 2019). Unlike these works that mainly focus on the performance of agents in a fixed team, we focus on the generalization ability of agents over different or even unseen teams. Also, these works mainly try to increase the mutual information between two variables to enhance the dependency between variables. However, in our work, we try to decrease the mutual information between the agent's policy and team-related information to reduce the dependency between these two variables and avoid overfitting. Being an exception, PMIC (Li et al., 2022) maximizes the MI associated with the superior trajectories and minimizes the MI associated with the inferior trajectories at the same time. However, it still focuses on the training of a fixed team, while our work focuses on the training of a dynamic team and the generalization over unseen teams.

## 3 Background

In this paper, we formulate cooperative MARL with dynamic team composition as a multi-agent Markov decision process (MMDP) with entities. MMDP with entities can be defined by a tuple $\langle \mathcal{E}, S, A, U, P, r, \gamma \rangle$. $\mathcal{E}$ is the set of entities in the environment, $S$ is the set of states, and each state $s$ is composed by the state of each entity $s = \{s_e\}$. It is worth noting that except agents $a \in A$, there are also other entities in the environments (e.g., landmarks, obstacles, agents with fixed behavior). $U = U_1 \times \cdots \times U_{|A|}$ is the joint action space, where $U_i$ is the individual action space for each agent $i$. For the rigorousness of proof, we assume full observability such that at each state $s \in S$, each agent $i$ receives state $s$, chooses an action $u_i \in U_i$, and all actions form a joint action $\boldsymbol{u} \in U$. The state transitions to the next state $s'$ upon $\boldsymbol{u}$ according to the transition function $P(s'|s, \boldsymbol{u}) : S \times U \times S \to [0, 1]$, and all agents receive a shared reward $r(s, \boldsymbol{u}) : S \times U \to \mathbb{R}$. The objective is to learn an individual policy $\pi_i(u_i|s)$ for each agent such that they can cooperate to maximize the expected cumulative discounted return, $\mathbb{E}[\sum_{t=0}^{\infty} \gamma^t r_t]$, where $\gamma \in [0, 1)$ is the discount factor. In CTDE, from a centralized perspective, a group of local policies can be viewed as a joint policy $\pi_{jt}(\boldsymbol{u}|s)$. For this joint policy, we can define the joint state-action value function $Q_{jt}(s_t, \boldsymbol{u}_t) = \mathbb{E}_{s_{t+1:\infty}, \boldsymbol{u}_{t+1:\infty}}[\sum_{k=0}^{\infty} \gamma^t r_{t+k}|s_t, \boldsymbol{u}_t]$. Note that although we assume full observability for the rigorousness of proof, we use the trajectory of each agent $\tau_i \in \mathcal{T}_i : (Y \times U_i)^*$ to replace state $s$ for each agent to settle the partial observability in practice, where $Y$ is the observation space.

Since we are discussing dynamic team composition in this paper, we further denote $s_i^+$ as team-unrelated information for agent $i$ (agent's own information), and use $s_i^-$ to denote team-related information that varies along team composition (e.g., information of other agents, landmarks, obstacles). Although we assume full observability for each agent (i.e., $s$ is the same for all agents), $s_i^+$ and $s_i^-$ can be different as the circumstance of each agent per se is different. One can easily conclude that $s = \{s_i^+, s_i^-\}$ for each agent $i$.

## 4 Method

In this section, we present our method, MIPI, as follows. In Section 4.1, we introduce the mutual information (MI) augmented objective for regularizing the reliance on $s_i^-$ for each agent. However, due to the existence of the dynamic marginal distribution, direct optimization on this objective can be intractable in practice. Therefore, in Section 4.2, we first discuss a multi-agent policy iteration with a fixed marginal distribution and prove its convergence and optimality. Then, in Section 4.3, we discuss how to use an imaginary team composition distribution to achieve an approximate dynamic marginal distribution and how to use the Blahut–Arimoto algorithm (Cover, 1999) to optimize the corresponding objective. Finally, in Section 4.4, we summarize the learning framework of MIPI.

### 4.1 MI-Augmented Objective

The learning objective of standard MARL can be formulated as follows,

$$\arg\max_{\pi_{jt}} \mathbb{E}_{\rho(s_0), \pi_{jt}, P} \left[ \sum_{t=0}^{T} \gamma^t r(s_t, \boldsymbol{u}_t) \right], \tag{1}$$

where $\rho(\boldsymbol{s}_0)$ is the distribution of initial state. We can further rewrite it as:

$$\arg\max_{\pi_{\mathrm{jt}}} \sum_{t=0}^{T} \mathbb{E}_{\rho_{\pi_{\mathrm{jt}}}(\boldsymbol{s}_t)} \left[ \mathbb{E}_{\pi_{\mathrm{jt}}(\boldsymbol{u}_t|\boldsymbol{s}_t)} \left[ \gamma^t r(\boldsymbol{s}_t, \boldsymbol{u}_t) \right] \right], \tag{2}$$

where $\rho_{\pi_{\mathrm{jt}}}(\boldsymbol{s}_t)$ is the the marginal distribution over states at timestep $t$. Recall that the conditional mutual information between $x$ and $y$ given $z$ can be expressed as follows,

$$\mathbf{MI}(x; y|z) = \mathbb{E}_{p(y,z)} \left[ \mathbb{E}_{p(x|y,z)} \left[ \log \frac{p(x|y,z)}{p(x|z)} \right] \right].$$

The conditional mutual information can be used to measure the dependency between variable $x$ and $y$ with $z$ given. Therefore, as our goal is to reduce the reliance of $\pi_i(u_i|\boldsymbol{s}) = \pi_i(u_i|s_i^+, s_i^-)$ on $s_i^-$, we can formulate conditional mutual information as follows,

$$\mathbf{MI}(u_i; s_i^-|s_i^+) = \mathbb{E}_{\rho(s_i^+, s_i^-)} \left[ \mathbb{E}_{\pi_i(u_i|s_i^+, s_i^-)} \left[ \log \frac{\pi_i(u_i|s_i^+, s_i^-)}{\hat{\pi}_i(u_i|s_i^+)} \right] \right]$$

$$= \mathbb{E}_{\rho(s_i^+), \rho(s_i^-|s_i^+)} \left[ \mathbb{E}_{\pi_i(u_i|\boldsymbol{s})} \left[ \log \frac{\pi_i(u_i|\boldsymbol{s})}{\pi_i(u_i|s_i^+)} \right] \right],$$

where $\pi_i(u_i|s_i^+) = \sum_{s_i^-} \rho(s_i^-|s_i^+)\pi_i(u_i|s_i^+, s_i^-)$. Incorporating the conditional mutual information of all agents into the standard MARL objective, we now have the MI-augmented objective used in this paper:

$$\arg\max_{\pi_{\mathrm{jt}}} \sum_{t=0}^{T} \mathbb{E}_{\rho_{\pi_{\mathrm{jt}}}(\boldsymbol{s}_t)} \left[ \mathbb{E}_{\pi_{\mathrm{jt}}(\boldsymbol{u}_t|\boldsymbol{s}_t)} \left[ \gamma^t \left( r(\boldsymbol{s}_t, \boldsymbol{u}_t) - \alpha \sum_i \log \frac{\pi_i(u_{i,t}|\boldsymbol{s}_t)}{\pi_i(u_{i,t}|s_{i,t}^+)} \right) \right] \right] \tag{3}$$

$$\text{s.t.} \quad \pi_i(u_{i,t}|s_{i,t}^+) = \sum_{s_{i,t}^-} \rho_{\pi_{\mathrm{jt}}}(s_{i,t}^-|s_{i,t}^+)\pi_i(u_{i,t}|s_{i,t}^+, s_{i,t}^-), \tag{4}$$

where the coefficient $\alpha$ is used to determine the trade-off between maximizing global reward and minimizing mutual information.

## 4.2 Multi-Agent Policy Iteration with a Fixed Marginal Distribution

As we can see in (4), the optimization of (3) is highly coupled with a dynamic marginal distribution $\pi_i(u_{i,t}|s_{i,t}^+)$. What's even worse is, this marginal distribution is determined by $\rho_{\pi_{\mathrm{jt}}}(s_{i,t}^-|s_{i,t}^+)$, which is related to $\pi_{\mathrm{jt}}$, making the optimization problem even harder. However, one may notice that, if such a marginal distribution $\pi_i(u_{i,t}|s_{i,t}^+)$ is given and fixed, this problem becomes much easier and can be solved in an off-policy manner. Therefore, in this section, we introduce multi-agent policy iteration with a fixed marginal distribution and prove its convergence and optimality, and in the next section, we discuss how to approximate the dynamic marginal distribution by integrating constraints similar to (4) into this multi-agent policy iteration. First, let us define the joint value function $V_{\mathrm{jt}}$ and joint state-action value function $Q_{\mathrm{jt}}$ as follows,

$$V_{\mathrm{jt}}^{\pi_{\mathrm{jt}}}(\boldsymbol{s}) = \mathbb{E}_{\pi_{\mathrm{jt}}} \left[ \sum_t \gamma^t \left( r(\boldsymbol{s}_t, \boldsymbol{u}_t) - \alpha \sum_i \log \frac{\pi_i(u_{i,t}|\boldsymbol{s}_t)}{\pi_i(u_{i,t}|s_{i,t}^+)} \right) | \boldsymbol{s}_0 = \boldsymbol{s} \right]$$

$$Q_{\mathrm{jt}}^{\pi_{\mathrm{jt}}}(\boldsymbol{s}, \boldsymbol{u}) = r(\boldsymbol{s}, \boldsymbol{u}) + \gamma \mathbb{E}_{\boldsymbol{s}' \sim P} \left[ V_{\mathrm{jt}}(\boldsymbol{s}') \right],$$

where $\pi_i(u_{i,t}|s_{i,t}^+)$ is the fixed marginal distribution for each agent $i$. With the above definition, we can further deduce that:

$$V_{\mathrm{jt}}^{\pi_{\mathrm{jt}}}(\boldsymbol{s}) = \mathbb{E}_{\pi_{\mathrm{jt}}} \left[ Q_{\mathrm{jt}}^{\pi_{\mathrm{jt}}}(\boldsymbol{s}, \boldsymbol{u}) - \alpha \sum_i \log \frac{\pi_i(u_i|\boldsymbol{s})}{\pi_i(u_i|s_i^+)} \right].$$

We can then define the joint policy evaluation operator as

$$\Gamma_{\pi_{\mathrm{jt}}} Q_{\mathrm{jt}}(\boldsymbol{s}, \boldsymbol{u}) := r(\boldsymbol{s}, \boldsymbol{u}) + \gamma \mathbb{E}_{\boldsymbol{s}'}[V_{\mathrm{jt}}(\boldsymbol{s}')] \tag{5}$$

and have the following lemma.

**Lemma 1** (**Joint Policy Evaluation**). *Consider the modified Bellman backup operator* $\Gamma_{\pi_{jt}}$ (5) *and a mapping* $Q^0_{jt} : S \times U \to \mathbb{R}$ *with* $|U| < \infty$, *and define* $Q^{k+1}_{jt} = \Gamma_{\pi_{jt}} Q^k_{jt}$. *Then, the sequence* $Q^k_{jt}$ *will converge to the joint Q-function of* $\pi_{jt}$ *as* $k \to \infty$.

*Proof.* See Appendix A.1. □

Using Lemma 1, we can get $Q_{jt}$ for any joint policy $\pi_{jt}$. However, it is hard for us to use $Q_{jt}$ for individual policy improvement. To solve this problem, many value decomposition methods choose to factorize the joint state-action value function $Q_{jt}$ into the utility function $Q_i$ of each agent and use $Q_i$ to guide the individual policy improvement of $\pi_i$. In this paper, we factorize the joint state-action value function into the following form, which is shared by many value decomposition methods (Zhang et al., 2021; Su and Lu, 2022; Wang et al., 2023):

$$Q^{\pi_{jt}}_{jt}(\boldsymbol{s}, \boldsymbol{u}) = \sum_i w_i(\boldsymbol{s}) * Q^{\pi_i}_i(\boldsymbol{s}, u_i) + b(\boldsymbol{s}). \tag{6}$$

After the evaluation of the joint policy and the decomposition of $Q_{jt}$, we construct the following optimization problem for individual policy improvement.

$$\pi^{\text{new}}_i = \arg\max_{\pi'_i} \mathbb{E}_{\pi'_i} \left[ Q^{\pi^{\text{old}}_i}_i(\boldsymbol{s}, u_i) - \alpha \log \frac{\pi'_i(u_i|\boldsymbol{s})}{\pi_i(u_i|s^+_i)} \right] \tag{7}$$

Based on the above optimization problem, we have the following lemma for individual policy improvement.

**Lemma 2** (**Individual Policy Improvement**). *Let* $\pi^{\text{new}}_i$ *be the optimizer of the maximization problem in* (7). *Then, we have* $Q^{\pi^{\text{new}}_{jt}}_{jt}(\boldsymbol{s}, \boldsymbol{u}) \geq Q^{\pi^{\text{old}}_{jt}}_{jt}(\boldsymbol{s}, \boldsymbol{u})$ *for all* $(\boldsymbol{s}, \boldsymbol{u}) \in |S| \times |U|$ *with* $|U| < \infty$, *where* $\pi^{\text{old}}_{jt}(\boldsymbol{u}|\boldsymbol{s}) = \prod_i \pi^{\text{old}}_i(u_i|\boldsymbol{s})$ *and* $\pi^{\text{new}}_{jt}(\boldsymbol{u}|\boldsymbol{s}) = \prod_i \pi^{\text{new}}_i(u_i|\boldsymbol{s})$.

*Proof.* See Appendix A.2. □

Combining Lemma 1 and 2, we can have the following theorem which proves the convergence and optimality of multi-agent policy iteration with a fixed marginal distribution.

**Theorem 1** (**Multi-Agent Policy Iteration with a Fixed Marginal Distribution**). *For any joint policy* $\pi_{jt}$, *if we repeatedly apply joint policy evaluation and individual policy improvement. Then the joint policy* $\pi_{jt}(\boldsymbol{u}|\boldsymbol{s}) = \prod^n_{i=1} \pi_i(u_i|\boldsymbol{s})$ *will eventually converge to* $\pi^*_{jt}$, *such that* $Q^{\pi^*_{jt}}_{jt}(\boldsymbol{s}, \boldsymbol{u}) \geq Q^{\pi_{jt}}_{jt}(\boldsymbol{s}, \boldsymbol{u})$ *for all* $\pi_{jt}$, *assuming* $|U| < \infty$.

*Proof.* See Appendix A.3. □

### 4.3 Approximation for Dynamic Marginal Distribution

With the multi-agent policy iteration above, we can have $Q_{jt}$ and corresponding $Q_i$ for each agent, however, under a fixed marginal distribution. In this section, we discuss how to approximate the dynamic marginal distribution to decouple $\rho_{\pi_{jt}}(s^-_{i,t}|s^+_{i,t})$ from $\pi_{jt}$, and introduce the Blahut–Arimoto algorithm for the corresponding optimization.

Notice that the original objective (3) comes with a constraint (4). In Section 4.2, what we did is to remove this constraint for an easier optimization process. What we are going to do here, is to add a similar constraint back. First, consider the meaning of $\rho_{\pi_{jt}}(s^-_i|s^+_i)$, it represents the potential team composition given team-unrelated information. Therefore, inspired by REFIL (Iqbal et al., 2021), we randomly partition team composition under $\boldsymbol{s} = \{s^+_i, s^-_i\}$ into different subgroups, yielding a set of imaginary team compositions and corresponding imaginary distribution $\hat{\rho}(s^*_i|s^+_i)$ for imaginary team-related information. With this imaginary distribution, we can propose the approximate objective

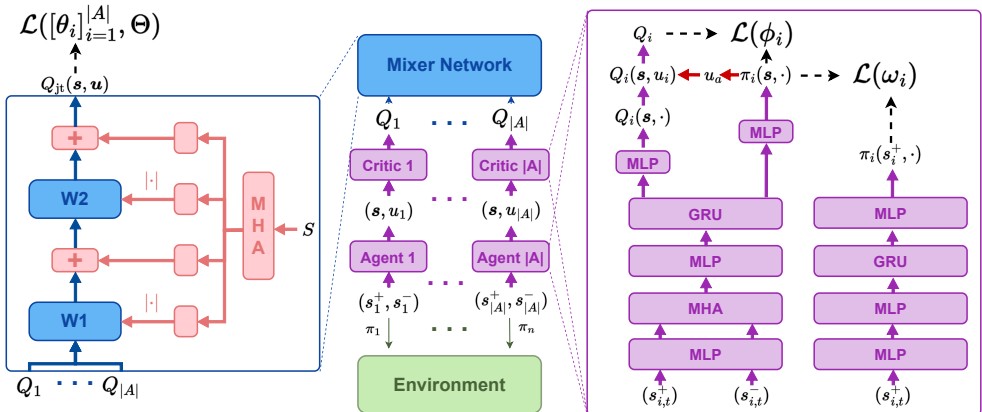

**Figure 1:** Learning framework of MIPI, where each agent $i$ has three modules: a utility function $Q_i(\boldsymbol{s}, u_i; \theta_i)$, a policy $\pi_i(u_i|\boldsymbol{s}; \phi_i)$, and a marginal policy $\pi_i(u_i|s_i^+; \omega_i)$.

for (3) as follows,

$$\arg\max_{\pi_{\text{jt}}} \sum_{t=0}^{T} \mathbb{E}_{\rho_{\pi_{\text{jt}}}(\boldsymbol{s}_t)} \left[ \mathbb{E}_{\pi_{\text{jt}}(\boldsymbol{u}_t|\boldsymbol{s}_t)} \left[ \gamma^t \Big( r(\boldsymbol{s}_t, \boldsymbol{u}_t) - \alpha \sum_i \log \frac{\pi_i(u_{i,t}|\boldsymbol{s}_t)}{\pi_i(u_{i,t}|s_{i,t}^+)} \Big) \right] \right] \tag{8}$$

$$\text{s.t.} \quad \pi_i(u_{i,t}|s_{i,t}^+) = \sum_{s_{i,t}^*} \hat{\rho}(s_{i,t}^*|s_{i,t}^+) \pi_i(u_{i,t}|s_{i,t}^+, s_{i,t}^*). \tag{9}$$

Therefore, we can have the approximate optimization problem for (7) as follows,

$$\pi_i^{\text{new}} = \arg\max_{\pi_i'} \mathbb{E}_{\pi_i'} \left[ Q_i^{\pi_i^{\text{old}}}(\boldsymbol{s}, u_i) - \alpha \log \frac{\pi_i'(u_i|\boldsymbol{s})}{\pi_i'(u_i|s_i^+)} \right] \tag{10}$$

$$\text{s.t.} \quad \pi_i'(u_i|s_i^+) = \sum_{s_i^*} \hat{\rho}(s_i^*|s_i^+) \pi_i'(u_i|s_i^+, s_i^*). \tag{11}$$

The objective above exhibits similarities with the rate-distortion problem (Cover, 1999), which could be solved using the Blahut–Arimoto algorithm. Although with the approximation above we break the convergence of Theorem 1, by using the Blahut–Arimoto algorithm we can have the following theorem, indicating the convergence of (10) as shown in Leibfried and Grau-Moya (2020).

**Theorem 2** (**Convergence of Constrained Individual Policy Improvement**). *The optimization problem induced by* (10) *can be solved by iterating in an alternate fashion through the following two equations:*

$$\pi_i^m(u_i|s_i^+) = \sum_{s_i^*} \hat{\rho}(s_i^*|s_i^+) \pi_i^m(u_i|s_i^+, s_i^*) \tag{12}$$

$$\pi_i^{m+1}(u_i|s_i^+, s_i^-) = \frac{\pi_i^m(u_i|s_i^+) \exp(Q_i(\boldsymbol{s}, u_i)/\alpha)}{\sum_{u_i} \pi_i^m(u_i|s_i^+) \exp(Q_i(\boldsymbol{s}, u_i)/\alpha)}, \tag{13}$$

*where $m$ refers to the iteration index. Denoting the total number of iterations as $M$, the presented scheme converges at a rate of $O(1/M)$ to an optimal policy $\pi_i^*$ for any given bounded utility function $Q_i$ and any initial policy $\pi_i^0$.*

*Proof.* See Appendix A.4. $\square$

## 4.4 MIPI Framework

In Section 4.2 and 4.3, we propose our learning algorithm in theory. In this section, we discuss how to implement our algorithm in practice, which can be summarized in Figure 1.

In MIPI, each agent has a utility function $Q_i(\boldsymbol{s}, u_i; \theta_i)$, a policy $\pi_i(u_i|\boldsymbol{s}; \phi_i)$, and a marginal policy $\pi_i(u_i|s_i^+; \omega_i)$. For joint policy evaluation, with the utilities of agents, we use a mixer network $\text{Mixer}(\cdot, \boldsymbol{s}; \Theta)$ to get the joint state-action value function $Q_{\text{jt}}$ as follows,

$$Q_{\text{jt}}(\boldsymbol{s}, \boldsymbol{u}) = \text{Mixer}([Q_i(\boldsymbol{s}, u_i; \theta_i)]_{i=1}^{|A|}, \boldsymbol{s}; \Theta) \tag{14}$$

$$= \sum_{i=1}^{|A|} w_i(\boldsymbol{s})Q_i(\boldsymbol{s}, u_i; \theta_i) + b(\boldsymbol{s}), \tag{15}$$

Where $w_i(\boldsymbol{s}) \geq 0$ is a positive weight used to linearly decompose $Q_{\text{jt}}$ with the IGM principle being preserved. Same as REFIL, $w_i(\boldsymbol{s})$ is computed via the attention mechanism to handle dynamic team composition. With $Q_{\text{jt}}$, we can update the utilities and the mixer network by minimizing the following TD error:

$$\mathcal{L}([\theta_i]_{i=1}^{|A|}, \Theta) = \mathbb{E}_{\mathcal{D}}\left[Q_{\text{jt}}(\boldsymbol{s}, \boldsymbol{u}) - \left(r(\boldsymbol{s}, \boldsymbol{u}) + \gamma\left(\hat{Q}_{\text{jt}}(\boldsymbol{s}', \boldsymbol{u}') - \alpha\sum_i^{|A|}\log\frac{\pi_i(u_i'|\boldsymbol{s}')}{\pi_i(u_i'|s_i'^+)}\right)\right)\right], \tag{16}$$

where $\mathcal{D}$ is the replay buffer, $\hat{Q}_{\text{jt}}$ is the target network and $u_i'$ is sampled from the current policy $\pi_i(u_i|\boldsymbol{s}; \phi_i)$ of each agent. To accelerate the training with a fixed set of team compositions, we also incorporate the same imaginary objective based on random sub-group partitioning as REFIL for joint policy evaluation.

As we described in Section 4.3, the constrained individual policy improvement is achieved via an iterative update of $\pi_i(u_i|\boldsymbol{s}; \phi_i)$ and $\pi_i(u_i|s_i^+; \omega_i)$. For $\pi_i(u_i|s_i^+; \omega_i)$, we update it via the maximum likelihood estimation:

$$\mathcal{L}(\omega_i) = \mathbb{E}_{\mathcal{D}}\left[\mathbb{E}_{s_i^* \sim \hat{\rho}(s_i^*|s_i^+), u_i \sim \pi_i(u_i|s_i^+, s_i^*)}\left[\log\pi_i(u_i|s_i^+)\right]\right]. \tag{17}$$

For $\pi_i(u_i|\boldsymbol{s}; \phi_i)$, we update it by minimizing the KL-divergence as follows,

$$\mathcal{L}(\phi_i) = \mathbb{E}_{\mathcal{D}}\left[\mathbb{E}_{u_i \sim \pi_i(u_i|\boldsymbol{s})}\left[\alpha\left(\log\frac{\pi_i(u_i|\boldsymbol{s})}{\pi_i(u_i|s_i^+)}\right) - Q_i(\boldsymbol{s}, u_i)\right]\right]. \tag{18}$$

## 5 Experiments

In this section, we evaluate MIPI in two different scenarios. One is a simple yet challenging matrix game, which we use to illustrate how mutual information may help to learn generalizable policies. Then, we evaluate MIPI in a complicated cooperative MARL scenario: StarCraft Multi-Agent Challenge (SMAC) (Samvelyan et al., 2019), comparing it against REFIL, AQMIX (Iqbal et al., 2021), CollaQ (Zhang et al., 2020) and MAPPO (Yu et al., 2022). More details about experiments, hyperparameters, and the learning curve of each algorithm are included in Appendix B and C. All results are presented using the mean and standard deviation of five runs with different random seeds.

### 5.1 An Illustrative Example: Matrix Game

We first use a matrix game to explain how mutual information works in solving generalization problems. In this game, we have two agents, and each of them can take two actions $\{0, 1\}$ and can take one of the two types $\{A, B\}$. During training, we train these two agents under team compositions $(A, B)$ and $(B, A)$, where the corresponding payoff matrices are shown in Figure 2(a) and 2(b). However, during evaluation, we test their performance on team composition $(B, B)$, and have Figure 2(c) as the payoff matrix, which is different from training scenarios.

As we can see in Figure 2(a), 2(a), 2(c), we have different optimal joint actions in different team compositions. However, there exists a generalizable joint action $(a_1 = 0, a_2 = 0)$ that can achieve consistent performance regardless of team compositions, even if it is not an optimal joint action in any team composition.

In Figure 2(d), we plot the evaluation results of three algorithms during training, which is evaluated on team composition $(B, B)$. These three algorithms are all the same except they receive different rewards: pure environmental reward, environmental reward combined with entropy, and environmental

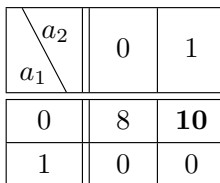 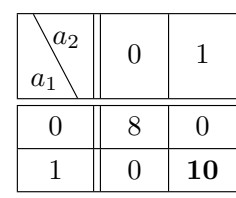 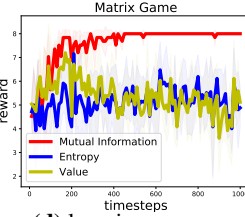

**(a)** payoff matrix: $(A, B)$  **(b)** payoff matrix: $(B, A)$  **(c)** payoff matrix: $(B, B)$  **(d)** learning curves

**Figure 2:** A matrix game with different team compositions: (a) (b) (c) payoff matrices for different team compositions; (d) learning curves of different methods on team composition $(B, B)$.

**Table 1:** Final performance on all SMAC maps. MIPI outperforms REFIL, AQMIX, and CollaQ in 8 out of 9 evaluation maps. We bold the best mean performance for each map.

| Tasks \ #Agent | Algorithms | Training 3-5 | Evaluation 6 | Evaluation 7 | Evaluation 8 |
|---|---|---|---|---|---|
| SZ | MIPI | 0.659±0.02 | **0.453**±0.08 | **0.404**±0.062 | **0.276**±0.076 |
| | REFIL | **0.674**±0.038 | 0.441±0.103 | 0.352±0.078 | 0.236±0.103 |
| | AQMIX | 0.528±0.044 | 0.343±0.105 | 0.291±0.084 | 0.182±0.058 |
| | CollaQ | 0.588±0.03 | 0.366±0.086 | 0.314±0.076 | 0.198±0.097 |
| | MAPPO | 0.256±0.01 | 0.129±0.019 | 0.148±0.031 | 0.036±0.015 |
| CSZ | MIPI | 0.548±0.032 | **0.42**±0.102 | **0.297**±0.112 | **0.261**±0.09 |
| | REFIL | **0.568**±0.027 | 0.348±0.057 | 0.229±0.053 | 0.164±0.06 |
| | AQMIX | 0.509±0.054 | 0.323±0.096 | 0.216±0.101 | 0.152±0.071 |
| | CollaQ | 0.459±0.061 | 0.362±0.13 | 0.267±0.099 | 0.231±0.095 |
| | MAPPO | 0.248±0.037 | 0.12±0.029 | 0.06±0.028 | 0.054±0.013 |
| MMM | MIPI | 0.548±0.023 | 0.495±0.054 | **0.447**±0.041 | **0.467**±0.067 |
| | REFIL | **0.605**±0.057 | 0.437±0.118 | 0.329±0.171 | 0.224±0.163 |
| | AQMIX | 0.501±0.036 | 0.447±0.043 | 0.344±0.071 | 0.251±0.089 |
| | CollaQ | 0.589±0.027 | **0.513**±0.07 | 0.423±0.026 | 0.286±0.083 |
| | MAPPO | 0.289±0.097 | 0.32±0.102 | 0.25±0.063 | 0.275±0.098 |

reward combined with mutual information. As we can see in Figure 2(d), with the help of mutual information, agents are able to resist the temptation of overfitting to the specific team composition and learn behavior that can generalize across different team compositions.

## 5.2 StarCraft Micromanagement Tasks

### 5.2.1 Performance

Further, we evaluate MIPI on SMAC with the map designed by Iqbal et al. (2021). We customize three different types of scenarios (SZ, CSZ, and MMM) based on this map for our experiments. In SZ scenarios, agents can take two different unit types, in CSZ and MMM, agents can take three different unit types. During training, the maps randomly initialize 3-5 agents and the same number of enemies at the start of each episode. During the evaluation, we use 6-8 agents and 6-8 enemies. Results are shown in Table 1. In general, MIPI outperforms the baselines in **8 out of 9 evaluation scenarios**. When the evaluation scenario is similar to the training scenarios, the gap between MIPI and other baselines is relatively small, whereas, in the evaluation scenario that is very different from the training scenarios, the gap between MIPI and other baselines becomes larger. In terms of the training performance, REFIL achieves the best result in all three scenarios, as it does not consider the overfitting issue at all. However, the performance of MIPI is still at the same level as REFIL, which indicates that MIPI can achieve better zero-shot generalization without sacrificing the performance on the training set.

**Table 2:** Final performance on all SMAC maps. MIPI is compared with ablation baselines. We bold the best mean performance for each map.

| #Agent / Tasks | Algorithms | Training | Evaluation | | |
|---|---|---|---|---|---|
| | | 3-5 | 6 | 7 | 8 |
| SZ | MIPI | **0.659**±0.02 | **0.453**±0.08 | **0.404**±0.062 | **0.276**±0.076 |
| | Value | 0.621±0.042 | 0.336±0.075 | 0.275±0.114 | 0.154±0.052 |
| | Entropy | 0.105±0.035 | 0.024±0.011 | 0.015±0.011 | 0.01±0.01 |
| CSZ | MIPI | **0.548**±0.032 | **0.42**±0.102 | **0.297**±0.112 | **0.261**±0.09 |
| | Value | 0.542±0.059 | 0.368±0.083 | 0.207±0.076 | 0.172±0.112 |
| | Entropy | 0.316±0.04 | 0.237±0.051 | 0.076±0.041 | 0.066±0.041 |
| MMM | MIPI | **0.548**±0.023 | 0.495±0.054 | **0.447**±0.041 | **0.467**±0.067 |
| | Value | 0.545±0.048 | **0.505**±0.058 | 0.391±0.083 | 0.319±0.105 |
| | Entropy | 0.265±0.034 | 0.2±0.085 | 0.16±0.064 | 0.075±0.044 |

### 5.2.2 Ablation

Although MIPI uses the random sub-group partitioning as in REFIL, it is an actor-critic structure, whereas REFIL uses only a critic. Therefore, one may question whether the improved generalization of MIPI is due to the introduction of mutual information, or simply due to the introduction of the actor. To eliminate such a concern, we build two ablation baselines, Value and Entropy, where all other perspectives are the same as MIPI, except they use pure environmental reward and entropy-augmented reward, respectively. As we can see in Table 2, MIPI also outperforms these two baselines, which demonstrates the importance of MI-augmented reward in MIPI.

## 6  Conclusion

In this paper, we propose MIPI, an MI-regularized multi-agent policy iteration algorithm to improve the generalization ability of agents under unseen team compositions. We first prove the convergence and optimality of our algorithm given a fixed marginal distribution, then we propose to use an imaginary distribution to approximate the dynamic marginal distribution to better approximate the original objective and incorporate the Blahut–Arimoto algorithm into the multi-agent policy iteration to optimize this approximate objective. We evaluate our algorithm in complex coordination scenarios, SMAC, and demonstrate that MIPI can achieve better zero-shot generalization results, without sacrificing the performance on the training set.

One potential limitation of this work is the introduction of approximation distorts the original mutual information augmented objective and breaks the convergences of multi-agent policy iteration. One possible solution to this problem is to seek alternative solutions using on-policy optimization methods (Schulman et al., 2015, 2017; Grudzien et al., 2022) to optimize the augmented objective.

## Acknowledgments and Disclosure of Funding

This work was supported in part by NSF China under grant 62250068 and Tencent. The authors would like to thank the anonymous reviewers for their valuable comments.

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

# A Proofs

## A.1 Proof of Lemma 1

**Lemma 1** (**Joint Policy Evaluation**). *Consider the modified Bellman backup operator $\Gamma_{\pi_{jt}}$ (5) and a mapping $Q_{jt}^0 : S \times U \to \mathbb{R}$ with $|U| < \infty$, and define $Q_{jt}^{k+1} = \Gamma_{\pi_{jt}} Q_{jt}^k$. Then, the sequence $Q_{jt}^k$ will converge to the joint Q-function of $\pi_{jt}$ as $k \to \infty$.*

*Proof.* First, define the augmented reward[2] as:

$$r_{\pi_{jt}}(\boldsymbol{s}, \boldsymbol{u}) := r(\boldsymbol{s}, \boldsymbol{u}) - \alpha \, \mathbb{E}_{\boldsymbol{s}'} \left[ \mathbb{E}_{\pi_{jt}} \left[ \sum_i \log \frac{\pi_i(u_i | \boldsymbol{s}')}{\pi_i(u_i | s_i'^+)} \right] \right].$$

Then, rewrite the update rule as:

$$Q_{jt}(\boldsymbol{s}, \boldsymbol{u}) \leftarrow r_{\pi_{jt}}(\boldsymbol{s}, \boldsymbol{u}) + \gamma \, \mathbb{E}_{\boldsymbol{s}', \boldsymbol{u}' \sim \pi_{jt}}[Q_{jt}(\boldsymbol{s}', \boldsymbol{u}')].$$

Last, apply the standard convergence results for policy evaluation (Sutton and Barto, 2018). □

## A.2 Proof of Lemma 2

**Lemma 2** (**Individual Policy Improvement**). *Let $\pi_i^{\text{new}}$ be the optimizer of the maximization problem in (7). Then, we have $Q_{jt}^{\pi_{jt}^{\text{new}}}(\boldsymbol{s}, \boldsymbol{u}) \geq Q_{jt}^{\pi_{jt}^{\text{old}}}(\boldsymbol{s}, \boldsymbol{u})$ for all $(\boldsymbol{s}, \boldsymbol{u}) \in |S| \times |U|$ with $|U| < \infty$, where $\pi_{jt}^{\text{old}}(\boldsymbol{u}|\boldsymbol{s}) = \prod_i \pi_i^{\text{old}}(u_i|\boldsymbol{s})$ and $\pi_{jt}^{\text{new}}(\boldsymbol{u}|\boldsymbol{s}) = \prod_i \pi_i^{\text{new}}(u_i|\boldsymbol{s})$.*

*Proof.* As $\pi_i^{\text{new}}$ optimizes (7), we can have:

$$\mathbb{E}_{\pi_i^{\text{new}}} \left[ Q_i^{\pi_i^{\text{old}}}(\boldsymbol{s}, u_i) - \alpha \log \frac{\pi_i^{\text{new}}(u_i|\boldsymbol{s})}{\pi_i(u_i|s_i^+)} \right] \geq \mathbb{E}_{\pi_i^{\text{old}}} \left[ Q_i^{\pi_i^{\text{old}}}(\boldsymbol{s}, u_i) - \alpha \log \frac{\pi_i^{\text{old}}(u_i|\boldsymbol{s})}{\pi_i(u_i|s_i^+)} \right]. \quad (19)$$

Since we assume that:

$$Q_{jt}^{\pi_{jt}}(\boldsymbol{s}, \boldsymbol{u}) = \sum_i w_i(\boldsymbol{s}) * Q_i^{\pi_i}(\boldsymbol{s}, u_i) + b(\boldsymbol{s}),$$

we can have:

$$\mathbb{E}_{\boldsymbol{u} \sim \pi_{jt}^{\text{new}}} \left[ Q_{jt}^{\pi_{jt}^{\text{old}}}(\boldsymbol{s}, \boldsymbol{u}) - \alpha \sum_i \log \frac{\pi_i^{\text{new}}(u_i|\boldsymbol{s})}{\pi_i(u_i|s_i^+)} \right]$$

$$= \mathbb{E}_{\boldsymbol{u} \sim \pi_{jt}^{\text{new}}} \left[ \sum_i w_i(\boldsymbol{s}) * Q_i^{\pi_i^{\text{old}}}(\boldsymbol{s}, u_i) + b(\boldsymbol{s}) - \alpha \sum_i \log \frac{\pi_i^{\text{new}}(u_i|\boldsymbol{s})}{\pi_i(u_i|s_i^+)} \right]$$

$$= \sum_i \mathbb{E}_{u_i \sim \pi_i^{\text{new}}} \left[ w_i(\boldsymbol{s}) * Q_i^{\pi_i^{\text{old}}}(\boldsymbol{s}, u_i) - \alpha \log \frac{\pi_i^{\text{new}}(u_i|\boldsymbol{s})}{\pi_i(u_i|s_i^+)} \right] + b(\boldsymbol{s})$$

$$\geq \sum_i \mathbb{E}_{u_i \sim \pi_i^{\text{old}}} \left[ w_i(\boldsymbol{s}) * Q_i^{\pi_i^{\text{old}}}(\boldsymbol{s}, u_i) - \alpha \log \frac{\pi_i^{\text{old}}(u_i|\boldsymbol{s})}{\pi_i(u_i|s_i^+)} \right] + b(\boldsymbol{s})$$

$$= \mathbb{E}_{\boldsymbol{u} \sim \pi_{jt}^{\text{old}}} \left[ Q_{jt}^{\pi_{jt}^{\text{old}}}(\boldsymbol{s}, \boldsymbol{u}) - \alpha \sum_i \log \frac{\pi_i^{\text{old}}(u_i|\boldsymbol{s})}{\pi_i(u_i|s_i^+)} \right]$$

$$= V_{jt}^{\pi_{jt}^{\text{old}}}(\boldsymbol{s}), \quad (20)$$

where the inequality is from plugging in (19).

---

[2]We assume $\pi_i(u_i|\boldsymbol{s})$ and $\pi_i(u_i|s_i^+)$ to be $\epsilon$-soft policy (Sutton and Barto, 2018) to avoid the log term being undefined.

Last, considering the modified Bellman equation, the following holds:

$$Q_{jt}^{\pi_{jt}^{old}}(\boldsymbol{s}, \boldsymbol{u}) = r(\boldsymbol{s}, \boldsymbol{u}) + \gamma \, \mathbb{E}_{\boldsymbol{s}'} \left[ V_{jt}^{\pi_{jt}^{old}}(\boldsymbol{s}') \right]$$

$$\leq r(\boldsymbol{s}, \boldsymbol{u}) + \gamma \, \mathbb{E}_{\boldsymbol{s}'} \left[ \mathbb{E}_{\boldsymbol{u}' \sim \pi_{jt}^{new}} \left[ Q_{jt}^{\pi_{jt}^{old}}(\boldsymbol{s}', \boldsymbol{u}') - \alpha \sum_i \log \frac{\pi_i^{new}(u_i'|\boldsymbol{s}')}{\pi_i(u_i'|s_i'^+)} \right] \right]$$

$$\vdots$$

$$\leq Q_{jt}^{\pi_{jt}^{new}}(\boldsymbol{s}, \boldsymbol{u}),$$

where we have repeatedly expanded $Q_{jt}^{\pi_{jt}^{old}}$ on the RHS by applying the modified Bellman equation and the inequality in (20). $\qquad \square$

## A.3 Proof of Theorem 1

**Theorem 1 (Multi-Agent Policy Iteration with a Fixed Marginal Distribution).** *For any joint policy $\pi_{jt}$, if we repeatedly apply joint policy evaluation and individual policy improvement. Then the joint policy $\pi_{jt}(\boldsymbol{u}|\boldsymbol{s}) = \prod_{i=1}^n \pi_i(u_i|\boldsymbol{s})$ will eventually converge to $\pi_{jt}^*$, such that $Q_{jt}^{\pi_{jt}^*}(\boldsymbol{s}, \boldsymbol{u}) \geq Q_{jt}^{\pi_{jt}}(\boldsymbol{s}, \boldsymbol{u})$ for all $\pi_{jt}$, assuming $|U| < \infty$.*

*Proof.* First, by Lemma 2, the sequence $\{\pi_{jt}^k\}$ monotonically improves with $Q_{jt}^{\pi_{jt}^{k+1}} \geq Q_{jt}^{\pi_{jt}^k}$. Since the augmented reward is bounded, then $Q_{jt}^{\pi_{jt}^k}$ is bounded. Thus, this sequence must converge to some $\pi_{jt}^*$. Then, at convergence, we have the following inequality:

$$\mathbb{E}_{\pi_i^*} \left[ Q_i^{\pi_i^*}(\boldsymbol{s}, u_i) - \alpha \log \frac{\pi_i^*(u_i|\boldsymbol{s})}{\pi_i(u_i|s_i^+)} \right] \geq \mathbb{E}_{\pi_i} \left[ Q_i^{\pi_i^*}(\boldsymbol{s}, u_i) - \alpha \log \frac{\pi_i(u_i|\boldsymbol{s})}{\pi_i(u_i|s_i^+)} \right], \forall \pi_i \neq \pi_i^*.$$

Using the same iterative argument as in the proof of Lemma 2, we get $Q_{jt}^{\pi_{jt}^*}(\boldsymbol{s}, \boldsymbol{u}) \geq Q_{jt}^{\pi_{jt}}(\boldsymbol{s}, \boldsymbol{u})$ for all $(\boldsymbol{s}, \boldsymbol{u}) \in |S| \times |U|$. That is, the state-action value of any other policy $\pi_{jt}$ is lower than or equal to that of the converged policy $\pi_{jt}^*$. Therefore, $\pi_{jt}^*$ is the optimal joint policy. $\qquad \square$

## A.4 Proof of Theorem 2

**Theorem 2 (Convergence of Constrained Individual Policy Improvement).** *The optimization problem in (10) can be solved by iterating in an alternate fashion through the following two equations:*

$$\pi_i^m(u_i|s_i^+) = \sum_{s_i^*} \hat{\rho}(s_i^*|s_i^+) \pi_i^m(u_i|s_i^+, s_i^*)$$

$$\pi_i^{m+1}(u_i|s_i^+, s_i^-) = \frac{\pi_i^m(u_i|s_i^+) \exp(Q_i(\boldsymbol{s}, u_i)/\alpha)}{\sum_{u_i} \pi_i^m(u_i|s_i^+) \exp(Q_i(\boldsymbol{s}, u_i)/\alpha)},$$

*where $m$ refers to the iteration index. Denoting the total number of iterations as $M$, the presented scheme converges at a rate of $O(1/M)$ to an optimal policy $\pi_i^*$ for any given bounded utility function $Q_i$ and any initial policy $\pi_i^0$.*

*Proof.* First, we notice that for a fixed $\pi_i(u_i|s_i^+, s_i^*)$, we can have its optimal marginal as constrained in (11):

$$\pi_i(u_i|s_i^+) = \sum_{s_i^*} \hat{\rho}(s_i^*|s_i^+) \pi_i(u_i|s_i^+, s_i^*).$$

Then, for a fixed marginal $\pi_i(u_i|s_i^+)$, we can have the optimal $\pi_i(u_i|s_i^+, s_i^*)$ by solving (10) via standard variational calculus:

$$\pi(u_i|s_i^+, s_i^-) = \frac{\pi(u_i|s_i^+) \exp(Q_i(\boldsymbol{s}, u_i)/\alpha)}{\sum_{u_i} \pi(u_i|s_i^+) \exp(Q_i(\boldsymbol{s}, u_i)/\alpha)}.$$

Lastly, with the above two equations, we can apply Theorem 1 in Leibfried and Grau-Moya (2020) to finish our proof. $\qquad \square$

# B  Experiment Settings and Implementation Details

## B.1  Matrix Game

In the matrix game, we use a learning rate of $3 \times 10^{-4}$ for all algorithms. For the algorithm that uses mutual information as the augmented reward, we set the number of Blahut–Arimoto iterations to 1. For algorithms that use mutual information and entropy as the augmented reward, we fix $\alpha$ as 0.5. The batch size used in the experiment is 64. Critics and polices used in the experiments consist of one hidden layer of 64 units with ELU non-linearity. For the mixer network, we use a hypernetwork similar to QMIX (Rashid et al., 2018), except no non-linearity is used. The environment and model are implemented in Python. All models are built by PyTorch and are trained via 1 Nvidia RTX 1060 GPU to conduct all the experiments. Each experiment takes roughly 1 hour.

## B.2  SMAC

In StarCraft II, we use a learning rate of $5 \times 10^{-4}$ for all algorithms. The structure of the critic network and the mixer network of MIPI are the same as REFIL (Iqbal et al., 2021) except no non-linearity is used in the mixer of MIPI. The number of Blahut–Arimoto iterations is set to 4 for MIPI in this experiment. The policy network of MIPI shares all layers with the critic network except the last layer of the policy network being a different fully-connected layer. The target networks will be updated once every 200 training episodes for all algorithms. The temperature parameters $\alpha$ and $\alpha_i$ are fixed as 0.03 in SZ and CSZ and fixed as 0.1 in MMM for MIPI and Entropy. For REFIL, AQMIX, and CollaQ, we use their default settings. For CollaQ, as the original implementation is based on a different SMAC environment where the entity-level observation is not available, we re-implement CollaQ with minimum changes to adapt the entity-level observation based on the framework provided in REFIL to ensure fairness of comparison. For MAPPO, as there is no published version of MAPPO for dynamic team compositions, we choose to implement MAPPO following Papoudakis et al. (2021), with additional attention modules used in the policy and the critic to handle dynamic team compositions. The environment and model are implemented in Python. All models are built by PyTorch and are trained via a mixture of 4 Nvidia A100, 4 RTX 3090, and 1 RTX 2080 TI GPUs to conduct all the experiments. Each experiment takes 6 to 32 hours depending on the algorithms and scenarios. Our implementation of MIPI is based on REFIL (Iqbal et al., 2021) with MIT license. It is worth noting that, although we assume full observability for the rigorousness of proof, the trajectory of each agent is used to replace state $s$ for each agent as input to settle the partial observability in all SMAC experiments.

## B.3  Resource Collection

In Resource Collection, we use a learning rate of $5 \times 10^{-4}$ for all algorithms. The structure of the critic network and the mixer network of MIPI are the same as REFIL (Iqbal et al., 2021) except no non-linearity is used in the mixer of MIPI. The number of Blahut–Arimoto iterations is set to 1 for MIPI in this experiment. The policy network of MIPI shares all layers with the critic network except the last layer of the policy network being a different fully-connected layer. The target networks will be updated once every 200 training episodes for all algorithms. The temperature parameters $\alpha$ and $\alpha_i$ are fixed as 0.05 in Resource Collection for MIPI. For REFIL, AQMIX, and CollaQ, we use their default settings. For CollaQ, as the original implementation is based on a different SMAC environment where the entity-level observation is not available, we re-implement CollaQ with minimum changes to adapt the entity-level observation based on the framework provided in REFIL to ensure fairness of comparison. For MAPPO, as there is no published version of MAPPO for dynamic team compositions, we choose to implement MAPPO following Papoudakis et al. (2021), with additional attention modules used in the policy and the critic to handle dynamic team compositions. The environment and model are implemented in Python. All models are built by PyTorch and are trained via 4 Nvidia RTX 3090 GPUs to conduct all the experiments. Each experiment takes roughly 20 hours. Our implementation of MIPI is based on REFIL (Iqbal et al., 2021) with MIT license. It is worth noting that, although we assume full observability for the rigorousness of proof, the trajectory of each agent is used to replace state $s$ for each agent as input to settle the partial observability in all SMAC experiments. As suggested by previous research (Liu et al., 2021; Shao et al., 2022), random sub-group partitioning does not work well in Resource Collection, therefore we choose not to use it for MIPI in this experiment.

# C Training performance on SMAC

In this section, we additionally provide the learning curves of all algorithms used in Section 5.2. As we can see from Figure 3, these algorithms achieve similar training performance, except Entropy.

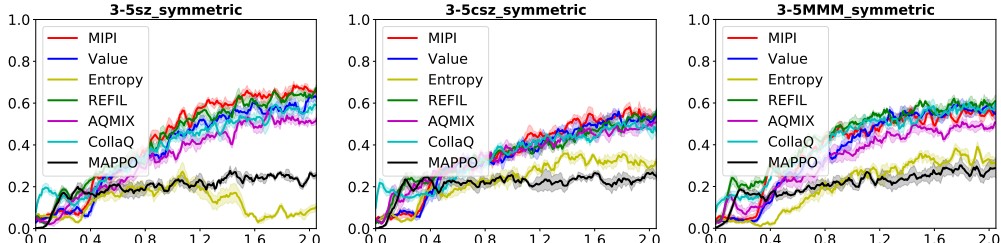

**Figure 3:** Learning curves of all the methods in SMAC, where the unit of x-axis is 1M timesteps and y-axis represents the win rate of each map.

# D More Experiments

## D.1 Resource Collection

In this section, We further evaluate MIPI on Resource Collection, which is a more challenging scenario in terms of the level of collaboration used by COPA (Liu et al., 2021). During training, the map randomly initializes 3-5 agents, and during the evaluation, we will have 6-8 agents. We plot the curve of training and evaluation performance in Figure 4. As we can see, MIPI outperforms the baselines by a large margin, which indicates that MIPI can also perform well in scenarios requiring strong collaboration.

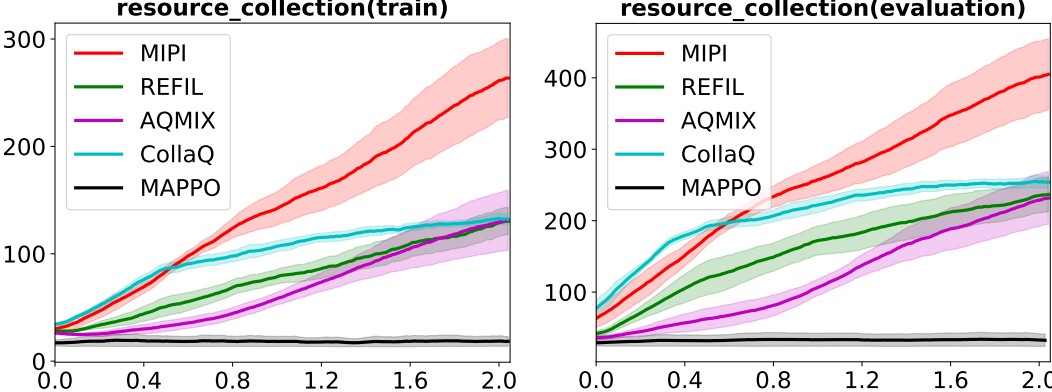

**Figure 4:** Learning curves of all the methods in Resource Collection, where the unit of x-axis is 1M timesteps and y-axis represents the return.

## D.2 Ablation on Alpha

In this section, We further include the ablation study on the impact of alpha. We train MIPI with different alpha on 3-5 agents scenarios and evaluate their performance on 6-8 agents scenarios. We use alpha=$\infty$ to represent the case where team-related information is completely removed (it is worth noting how this is different from actually set alpha=$\infty$ in MIPI). Results are summarized in Table 3. As we can see, unless alpha is set unreasonably, MIPI can always achieve better generalization ability without sacrificing the training performance. It's worth noting that alpha=$\infty$ outperforms Value here, which further indicates that reducing the dependency on team-related information promotes generalization, even when the team-related information is completely removed. However, this strategy is not widely effective and sacrifices the training performance too much in some cases (see MMM), which further leads to a decay in both training and evaluation. In contrast, our method uses alpha to control the degree of dependency on team-related information, which provides more flexibility.

**Table 3:** Final performance on all SMAC maps. MIPI is compared with the ablation baseline. We bold the best mean performance for each map.

| #Agent / Tasks | Alpha | Training 3-5 | Evaluation 6 | Evaluation 7 | Evaluation 8 |
|---|---|---|---|---|---|
| SZ | 0 (Value) | 0.621±0.042 | 0.336±0.075 | 0.275±0.114 | 0.154±0.052 |
| SZ | 0.01 | **0.672**±0.02 | 0.394±0.065 | 0.365±0.068 | 0.261±0.092 |
| SZ | 0.03 (MIPI) | 0.659±0.02 | **0.453**±0.08 | 0.404±0.062 | 0.276±0.076 |
| SZ | 0.05 | 0.643±0.02 | 0.447±0.062 | **0.408**±0.054 | **0.313**±0.069 |
| SZ | 0.1 | 0.475±0.073 | 0.277±0.125 | 0.26±0.093 | 0.146±0.075 |
| SZ | 0.5 | 0.175±0.053 | 0.056±0.015 | 0.129±0.026 | 0.043±0.021 |
| SZ | ∞ | 0.546±0.069 | 0.429±0.036 | 0.389±0.038 | 0.221±0.004 |
| CSZ | 0 (Value) | 0.542±0.059 | 0.368±0.083 | 0.207±0.076 | 0.172±0.112 |
| CSZ | 0.01 | **0.592**±0.02 | 0.378±0.033 | **0.364**±0.073 | **0.304**±0.056 |
| CSZ | 0.03 (MIPI) | 0.548±0.032 | **0.42**±0.102 | 0.297±0.112 | 0.261±0.09 |
| CSZ | 0.05 | 0.506±0.046 | 0.417±0.092 | 0.223±0.094 | 0.192±0.091 |
| CSZ | 0.1 | 0.344±0.05 | 0.218±0.16 | 0.113±0.079 | 0.098±0.086 |
| CSZ | ∞ | 0.506±0.076 | 0.368±0.064 | 0.309±0.056 | 0.27±0.07 |
| MMM | 0 (Value) | 0.545±0.048 | 0.505±0.058 | 0.391±0.083 | 0.319±0.105 |
| MMM | 0.05 | **0.59**±0.008 | **0.59**±0.053 | **0.526**±0.055 | 0.426±0.152 |
| MMM | 0.1 (MIPI) | 0.548±0.023 | 0.495±0.054 | 0.447±0.041 | **0.467**±0.067 |
| MMM | 0.5 | 0.277±0.042 | 0.158±0.094 | 0.18±0.105 | 0.139±0.056 |
| MMM | ∞ | 0.396±0.07 | 0.432±0.018 | 0.383±0.041 | 0.315±0.061 |

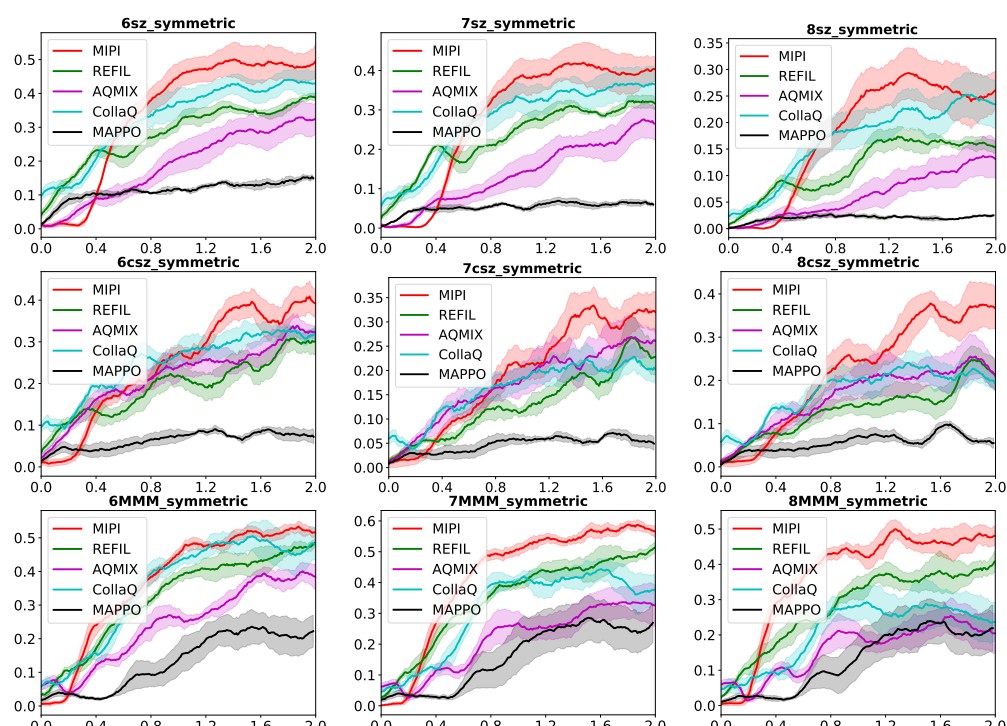

**Figure 5:** Learning curves of all the methods in SMAC, where the unit of x-axis is 1M timesteps and y-axis represents the win rate of each map.

## D.3  Performance on Higher Level Driver

When working on the follow-up project of this paper, we noticed that REFIL can achieve better generalization results in MMM with a higher-level NVIDIA driver without any code-level change.

The results are shown in Figure 5, where all algorithms are trained in a single platform that REFIL achieves better results. As we can see, although REFIL achieves better generalization results in some cases, MIPI can still outperform these baselines in terms of both speed and final performance by properly setting $\alpha$ and $\alpha_i$ (0.01 for sz, 0.015 for csz and 0.05 for MMM).

