# OpenReview forum: "Mutual-Information Regularized Multi-Agent Policy Iteration"
_NeurIPS.cc/2023/Conference — NeurIPS 2023 poster_

### Official Review · Reviewer_b1Qn · 2023-07-01

**Soundness:** 4 excellent
**Presentation:** 4 excellent
**Contribution:** 3 good
**Rating:** 6
**Confidence:** 3

**Summary:**

This paper propose a new method: LIMI, which utilizes mutual information as a regularizer to help MARL algorithms better face to the variation of team decomposition and generalize on unseen tasks. Both detailed theory proof and practical details are provided.

**Strengths:**

- The paper is well-written, both detailed proof and practical details are provided.
- Good empirical results on SMAC: better zero-shot generalization than other methods.

**Weaknesses:**

Main issue:
- More environments needed: SMAC is very sensitive to hyper-parameters. Therefore more results on other environments are needed to make the claims in the paper persuasive (e.g. smacv2, MAMuJuCo).

Minor issues:
- Where is the 'Ignore' of 'Learning to Ignore' in this paper?
- Two papers[1][2] should be cited as they used value decomposition in off-policy gradients method.

[1]  Dop: Off-policy multi-agent decomposed policy gradients\
[2]  Facmac: Factored multi-agent centralised policy gradients

**Questions:**

Will the code be released later?

**Limitations:**

No obvious negative societal impact.

---

> ### Author Rebuttal · Authors · 2023-08-09
>
> > More environments needed: SMAC is very sensitive to hyper-parameters. Therefore more results on other environments are needed to make the claims in the paper persuasive (e.g. smacv2, MAMuJuCo).
>
> Thanks for your valuable suggestion. We have carried out additional experiments on more scenarios to further verify our method. Please refer to the common response for more experiment results.
>
> However, as in the default setting of smacv2, team composition can only vary in terms of agent types, whereas in our smac experiments, following existing research [1][2][3], team composition can vary in both types and numbers, so we believe that our smac setting is more challenging than smacv2.
>
> Also, the released version of MAMuJoCo does not support the setting of dynamic team composition. Since the number of joints is fixed for each robot, changing the team that controls the robot is not available. Although we can reformulate MAMuJoCo completely to allow a robot controlled by different teams (for example, a 12 joints robot can be controlled by one 6-joint agent and two 3-joint agents, and it can also be controlled by four 3-joint agents), it requires a lot of efforts and is not easy to build such an environment from scratch. Thus, MAMuJoCo is currently out of consideration.
>
> > Where is the 'Ignore' of 'Learning to Ignore' in this paper?
>
> In our method, agents will learn to properly ignore team-related information to avoid overfitting. Such ignorance is achieved by reducing the mutual information between the agent's policy and team-related information.
>
> > Two papers[1] [2] should be cited as they used value decomposition in off-policy gradients method.
>
> Thanks for your comment. We will add corresponding citations in our related works section.
>
> > Will the code be released later?
>
> We will release our code once our paper is accepted.
>
> [1] Randomized entity-wise factorization for multi-agent reinforcement learning, Iqbal et al., ICML 2021.
>
> [2] Coach-player multi-agent reinforcement learning for dynamic team composition, Liu et al., ICML 2021
>
> [3] Self-Organized Group for Cooperative Multi-agent Reinforcement Learning, Shao et al., NIPS 2022.

---

> > ### Comment · Reviewer_b1Qn · 2023-08-14
> >
> > Thanks for the detailed explanation! As it addressed my concerns to some extent, I will keep the score. But I still strongly recommend author to mention more 'ignore' in the paper to avoid confusion.

---

> > > ### Author Response · Authors · 2023-08-15
> > > **Reply to Reviewer b1Qn**
> > >
> > > > But I still strongly recommend author to mention more 'ignore' in the paper to avoid confusion.
> > >
> > > We will address this problem in the later version of our paper. Thanks for your suggestion.
> > >
> > > We really appreciate your time and effort. Please feel free to ask us if you have additional questions, we will try our best to address your concerns during the discussion. Also, if you have additional suggestions, please do let us know so that we can further improve our paper. Again, thanks for reviewing our paper.

---

### Official Review · Reviewer_KNmz · 2023-07-03

**Soundness:** 3 good
**Presentation:** 2 fair
**Contribution:** 3 good
**Rating:** 5
**Confidence:** 3

**Summary:**

The paper presents a MARL algorithm that utilizes an SAC-based auxiliary mutual information reward regularizer to generalize learned decentralized multi-agent policies to different team compositions. The method has been evaluated against related baselines and in two simple and complex domains, demonstrating a relatively superior performance. At the current state of the paper, I have a few concerns and questions:

- There’s no clear definition of “team composition”, although understanding this terminology is pivotal for a reader. It’s not immediately clear what “composition” means in this context. I recommend adding a formal definition early on in the introduction.

- Related to the comment above, lack of such formal definitions makes it hard to understand the general, real-world motivation behind the solution presented here. Note that this is different from the technical motivation which is already provided. I recommend trying to give a real-world running example of a multi-agent scenario where such solution would be necessary.

- The approach and ideology presented here seems to be closely connected to InfoPG or MOA (Konan et al, 2022 and Jaques et al, 2019), and etc., where specifically in InfoPG (Konan et al, 2022) the notion of mutual information is leveraged through action-conditional policy distributions to help agents of a cooperative team ignore bad agents, i.e., bad information received from other agents for instance in a BGP problem. Yet I don’t see a mention of such works or discussion in this paper. For example, what problems can LIMI handle that InfoPG cannot? Therefore, the literature review under Section 2.3 seems weak and unconvincing.

[1] Konan, Sachin G., Esmaeil Seraj, and Matthew Gombolay. "Iterated Reasoning with Mutual Information in Cooperative and Byzantine Decentralized Teaming." International Conference on Learning Representations. 2021.

[2] Jaques, Natasha, et al. "Social influence as intrinsic motivation for multi-agent deep reinforcement learning." International conference on machine learning. PMLR, 2019.

- Directly adding a mutual information regularizer to the reward to “avoid relying too much on team information” or to minimize (as stated in the paper) sounds counter-intuitive for “cooperative” MARL where we do want agents to have higher mutual information of each other’s policy. Intuitively, the more an agent knows about its teammates actions and policies, the better decision it can make regarding its own action and policy. Also, methods like InfoPG ([1] above) indirectly maximize or minimize the mutual information between the policies of agents in a team through relying on positive or negative environment feedback and cooperativity of agents rather than manually forcing agents to “know less about each other” as suggested here. How do you justify this counter intuition and how do you balance?

- I might be wrong, but from Fig 1 and Eq 16-17, it seems like you would need access to s+ and s- (i.e., relevant and irrelevant information)? How would you identify/distinguish such information?

- Given that a major part of the contribution here is the zero-shot generalization and that LIMI doesn’t improve training performance significantly over existing benchmark, I suggest adding more results from more complex environments to showcase the potential.

- Please try to spell-out acronyms when you first define. It increases readability and understandability of your paper.

At current states I vote weak rejection, although the algorithm seems to be sound and working. I need to see more discussions and revisions as suggested above, as well as suggested by my fellow reviewers before changing my vote. I’d be happy to increase my score when authors satisfactorily addressed the comments.

**Strengths:**

See above

**Weaknesses:**

See above

**Questions:**

See above

**Limitations:**

One potential limitation is discussed. No potential societal impact are discussed. I think the provided discussions are relevant and satisfactory.

---

> ### Author Rebuttal · Authors · 2023-08-09
>
> > There’s no clear definition of “team composition”...
>
> As we described in Line[136-138], we assume each state $s$ can be decomposed into the state of entities $s_e$, and entities include agent entities and non-agent entities. So the term "team composition" here simply means the composition of agent entities, which may vary in terms of numbers and types. Thanks for pointing out this potentially misleading part, we will add corresponding texts in the later version of our paper for this.
>
> > I recommend trying to give a real-world running example of a multi-agent scenario where such solution would be necessary.
>
> Thanks for your comment. Including a running example surely eases the understanding of our paper. If we understand your question correctly, you are talking about the necessity of training agents that can generalize to dynamic team compositions. We can use the automated warehouse as an example, where it is very common for you to add more agents (more agents got purchased), or delete some agents (some agents got broken). Therefore, agents have to deal with different teams in the application scenario, which can not be fully covered during training. To solve this problem, training agents that can generalize to different teams becomes a necessary topic.
>
> > The approach and ideology presented here seems to be closely connected to InfoPG or MOA....
>
> Thanks for your comment. In our previous version of related works, we mainly focus on research where mutual information (and other informational measurements) does not simply serve as additional loss terms, but alters the form of policy iteration. However, we agree that we should also cover previous works like InfoPG and MOA, as they also account for a large part of MARL research where informational measurements are used. Therefore, we will add corresponding citations and texts in our related work section to emphasize the difference between our work and previous research in this line.
>
> We think the main difference between our work and InfoPG or MOA is that we focus on the generalization ability of agents over different teams, whereas both InfoPG and MOA focus on the performance of a fixed team. Also, InfoPG requires communication between agents to achieve action conditioned policy, whereas LIMI does not assume such a communication channel. In MOA, authors proposed to use agent modeling to avoid direct communication between agents. However, as the agent modeling module requires supervised training using the actions of other agents, therefore it may not work well in the generalization setting where the behavior of agents in unseen teams is not predictable.
>
> > Directly adding a mutual information regularizer... sounds counter-intuitive for “cooperative” MARL... Also, methods like InfoPG ([1] above) indirectly maximize or minimize the mutual information relying on environment feedback rather than manually forcing agents to “know less about each other”... How do you justify this counter intuition and how do you balance?
>
> Thanks for your comment. It is true that by knowing much about others, agents can make better decision and achieves a higher level of coordination. However, this is only true if we focus on just a single team (or a fixed set of teams) where agents can overfit each other at an arbitrary level without damaging the performance. When it comes to generalization, overfitting becomes a problem and leads to degenerated performance, which is common in deep learning algorithms [1]. Therefore, as we focus on the generalization ability of the agent's policy, we use MI as a regularizer to prevent agents from overfitting to the training set, which is similar to using the L2 norm to avoid the neural network from overfitting to the training set. Except, in our case, the overfitting is in terms of the dependency among agents rather than the parameters of the neural network. From this perspective, our method can be viewed as an instantiation of the regularization method in MARL, which becomes less counter-intuition.
>
> Again, same as those regularization methods, we have to balance the learning progress and the level of regularization. In our case, we use $\alpha$ to balance the trade-off between learning progress (reward) and regularization (MI).
>
> Also, as we focus on the generalization ability, where we may have to deal with unseen teams. It is hard to utilize the environmental feedback in the training set as it may vary a lot between the training team and the test team. However, extrapolating this environment feedback and using it for better regularization is indeed a promising direction, and we may work on this in our future work. But this is beyond the scope of this paper.
>
> > It seems like you would need access to s+ and s- (i.e., relevant and irrelevant information)? How would you identify/distinguish such information?
>
> As we mentioned in our paper Line[151-154], we will decompose observed information into team-unrelated information $s_{i}^{+}$, which includes information of agent per se and other non-agent information, and team-related information $s_{i}^{-}$, which includes information of other agents. To achieve this kind of decomposition, agents only need to distinguish different entities in the environment, which is commonly assumed in previous research, see [2] and [3].
>
> > Given that a major part of the contribution here is the zero-shot generalization and that LIMI doesn’t improve training performance significantly over existing benchmark, I suggest adding more results from more complex environments to showcase the potential.
>
> Please refer to the common response for more experiments.
>
> Additionally, we will correct the problem of acronyms.
>
> [1] An overview of overfitting and its solutions, Ying Xue, JPCS 2019.
>
> [2] Randomized entity-wise factorization for multi-agent reinforcement learning, Iqbal et al., ICML 2021.
>
> [3] Self-Organized Group for Cooperative Multi-agent Reinforcement Learning, Shao et al., NIPS 2022.

---

> > ### Comment · Reviewer_KNmz · 2023-08-15
> > **Thank you**
> >
> > Thank you to the authors for clarifications and providing further ablation results. I adjusted my scores accordingly.

---

> > > ### Author Response · Authors · 2023-08-15
> > > **Reply to Reviewer KNmz**
> > >
> > > We appreciate your time and effort. Your suggestions indeed help us improve the quality of our work. Again, thanks for reviewing our paper and raising the score.

---

### Official Review · Reviewer_KG7B · 2023-07-04

**Soundness:** 4 excellent
**Presentation:** 4 excellent
**Contribution:** 3 good
**Rating:** 6
**Confidence:** 3

**Summary:**

This paper proposes a method which use the mutual information as intrinsic reward to prevent the agents over-rely on its collaborators. Meanwhile, the proposed method can encourage the agents to learn robust policies in different team compositions. Moreover, this paper utilizes Blahut–Arimoto algorithm and an imaginary team composition distribution for optimization. The experiments show the effectiveness of the proposed method

**Strengths:**

1.	This paper proposes a novel method for improving multi-agent policy iteration.
2.	Experiment results on several scenarios in SCII show the efficiency of the proposed LIMI.


**Weaknesses:**

1.	Some important baselines are missed. Since the authors have mentioned them in the related work, the new published and common used baselines (e.g. MAPPO, HATRPO) should not be neglected.
2.	Performance on more challenging scenarios iof different benchmarks deserve to be verified.


**Questions:**

1.	Mutual information is widely used in MARL. What is the main difference or improvement of the proposed one compared with the related works[1-4]?
2.	How the model will perform compared with MAPPO?

[1] Celebrating diversity in shared multi-agent reinforcement learning. 2021.\
[2] Influence-Based Multi-Agent Exploration. 2019.\
[3] Social influence as intrinsic motivation for multiagent deep reinforcement learning. 2019\
[4] The emergence of individuality. 2021.


**Limitations:**

No potential negative societal impact.

---

> ### Author Rebuttal · Authors · 2023-08-09
>
> > Some important baselines are missed. Since the authors have mentioned them in the related work, the new published and common used baselines (e.g. MAPPO, HATRPO) should not be neglected.
>
> > Performance on more challenging scenarios of different benchmarks deserve to be verified.
>
> > How the model will perform compared with MAPPO?
>
> Thanks for your valuable suggestion. We have carried out additional experiments on more challenging scenarios and included MAPPO as a baseline in our experiments. Please refer to the common response for more experiment results.
>
> However, as you may have noticed, there is no published version of MAPPO for dynamic team compositions. We choose to implement MAPPO following [1], with additional attention modules used in the policy and the critic to handle dynamic team compositions as our baseline in our experiments.
>
> > Mutual information is widely used in MARL. What is the main difference or improvement of the proposed one compared with the related works[1-4]?
>
> There is a key difference between our work and other MARL methods using mutual information. In our work, we aim to **DECREASE** the mutual information between the agent's policy and team-related information to reduce the dependency between these two variables and avoid overfitting. In other works, they mainly try to **INCREASE** the mutual information between two variables to enhance the dependency between variables. Of course, there is no right or wrong here, we just try to use mutual information to solve different problems.
>
> [1] Benchmarking multi-agent deep reinforcement learning algorithms in cooperative tasks, Papoudakis et al., NIPS 2021.

---

> > ### Comment · Reviewer_KG7B · 2023-08-21
> >
> > I would like to thank the authors for their response and additional experiments.  I will keep my score as 6.

---

> > > ### Author Response · Authors · 2023-08-21
> > > **Reply to Reviewer KG7B**
> > >
> > > We appreciate your time and effort, thanks for reviewing our paper and providing useful suggestions.

---

### Official Review · Reviewer_NEb7 · 2023-07-09

**Soundness:** 2 fair
**Presentation:** 2 fair
**Contribution:** 2 fair
**Rating:** 6
**Confidence:** 4

**Summary:**

To address the issues of overfitting in Multi-Agent Reinforcement Learning (MARL) and dynamic team formation, this paper proposes a new approach of using mutual information regularization as intrinsic rewards to prevent individual policies from excessively relying on team-related information. Experimental results demonstrate that the proposed method outperforms relevant baselines on several SMAC tasks.

**Strengths:**

1. This paper is easy to read.
2. The method proposed in this paper, which utilizes mutual information to address the dynamic team formation problem, appears to be novel. However, I have some questions that need to be addressed by the authors.

**Weaknesses:**

1. This paper aims to reduce the reliance of agents on team-related information in decision-making. However, one potential issue may exsit with this approach. It may result in fewer cooperative behaviors among agents, leading to a greater emphasis on individual performance than team collaboration. As a result, this method may not perform well in tasks that require high levels of cooperation. I suggest that the authors evaluate their algorithm on tasks provided by COPA, which require strong collaboration.

2. The introduction of mutual information constraints introduces some human knowledge, as it requires decomposing the observed information into team-related and unrelated components.

3. The paper lacks some related works. In section 2.3 "Information-Theoretic Principles in RL," the authors should provide a more extensive overview of relevant MI-based MARL literature [1-10], which is more closely tied to collaboration.

---

[1] Maven: Multi-agent variational exploration.

[2] A Variational Approach to Mutual Information-Based Coordination for Multi-Agent Reinforcement Learning.

[3] An Adaptive Entropy-Regularization Framework for Multi-Agent Reinforcement Learning.

[4] Social Influence as Intrinsic Motivation for Multi-Agent Deep Reinforcement Learning.

[5] Influence-Based Multi-Agent Exploration.

[7] Episodic multi-agent reinforcement learning with curiosity-driven exploration.

[8] The Emergence of Individuality.

[9] PMIC: Improving Multi-Agent Reinforcement Learning with Progressive Mutual Information Collaboration.

[10] Celebrating Diversity in Shared Multi-Agent Reinforcement Learning.

**Questions:**

1. Could the authors perform algorithm comparisons on tasks that require higher levels of collaboration?
2. What would be the impact on performance if the team-related information is completely removed in SMAC tasks?
3. are the network architecture and basic hyperparameters of LIMI the same as those of the baselines?
4. Can the authors provide experimental analyses regarding the hyperparameter alpha? Additionally, what are the specific settings of alpha for each task?

**Limitations:**

See Weaknesses.

---

> ### Author Rebuttal · Authors · 2023-08-09
>
> > This paper aims to reduce the reliance of agents on team-related information in decision-making. However, one potential issue may exist with this approach. It may result in fewer cooperative behaviors among agents, leading to a greater emphasis on individual performance than team collaboration. As a result, this method may not perform well in tasks that require high levels of cooperation. I suggest that the authors evaluate their algorithm on tasks provided by COPA, which require strong collaboration.
>
> > Could the authors perform algorithm comparisons on tasks that require higher levels of collaboration?
>
> > What would be the impact on performance if the team-related information is completely removed in SMAC tasks?
>
> > Can the authors provide experimental analyses regarding the hyperparameter alpha?
>
> Thanks for your valuable suggestion. We have carried out additional experiments on the task that requires strong collaboration and ablation studies for the questions above. Please refer to the common response for more experiment results. Additionally, we would like to point out that reducing the reliance on team-related information will not lead to a greater emphasis on individual performance than on team collaboration. This is because, in cooperative MARL problems, agents can only receive a joint reward signal which reflects the joint performance of all agents. There is no individual reward signal for agents to optimize. Therefore, even if we reduce the reliance on team-related information, they will still be optimizing the joint reward, which is about the performance of the whole team. The reason to reduce the reliance of agents on team-related information is just to prevent agents from overfitting to team compositions used as the training set.
>
> > The introduction of mutual information constraints introduces some human knowledge, as it requires decomposing the observed information into team-related and unrelated components.
>
> Thanks for pointing this out. As we mentioned in our paper Line [151-154], we will decompose observed information into team-unrelated information $s_{i}^{+}$, which includes information of agent per se and other non-agent information, and team-related information $s_{i}^{-}$, which includes information of other agents. To achieve this kind of decomposition, agents only need to distinguish different entities in the environment, which is commonly assumed in previous research, see [1] and [2].
>
> However, your comment still makes sense in scenarios where such decomposition is not trivial. For example, when we have an image as the agent's input. In such cases, we can further relax the constraint of information decomposition, where team-unrelated information $s_{i}^{+}$ will only include information of agent per se, and team-related information $s_{i}^{-}$ will include all other information. With such a decomposition, agents only need to distinguish themselves from other information, which is much easier than the previous one. We use this setting in our Resource Collection experiment, where $s_{i}^{+}$ of LIMI is just the information of the agent itself, and $s_{i}^{-}$ includes information about other agents and landmarks. As suggested by the result, LIMI can still achieve better performance than other baselines with this relaxed way of information decomposition. This kind of information decomposition is also consistent with our setting in SMAC, as there are no non-agent entities in SMAC. So, there is no need to redo the experiments in SMAC.
>
> > The paper lacks some related works. In section 2.3 "Information-Theoretic Principles in RL," the authors should provide a more extensive overview of relevant MI-based MARL literature [1-10], which is more closely tied to collaboration.
>
> Thanks for your comment. We will add corresponding discussions of these papers in our related works section. In our previous version of related works, we mainly focus on research where mutual information (and other informational measurements) does not simply serve as additional loss terms, but alters the form of policy iteration. But we will surely add more texts and citations in the later version of our paper based on your recommendations, as these works also account for a large part of MARL research where informational measurements are used. Unlike these works that mainly focus on the performance of agents in a fixed team, we focus on the generalization ability of agents over different or even unseen teams. Also, these works mainly try to **INCREASE** the mutual information between two variables to enhance the dependency between variables (except PMIC). However, in our work, we try to **DECREASE** the mutual information between the agent's policy and team-related information to reduce the dependency between these two variables and avoid overfitting.
>
> > Are the network architecture and basic hyperparameters of LIMI the same as those of the baselines?
>
> Yes, as we implement our algorithm based on REFIL. The network architecture of the agent and mixer, also the basic hyperparameters of LIMI are the same as REFIL. Except as required by our theory, we remove the non-linearity of the hypernetwork produced by the mixer.
>
> > Additionally, what are the specific settings of alpha for each task?
>
> As we mentioned in Line [468-469] in Appendix, $\alpha$ and $\alpha_i$ are fixed as 0.03 in SZ and CSZ and fixed as 0.1 in MMM for LIMI.
>
> [1] Randomized entity-wise factorization for multi-agent reinforcement learning, Iqbal et al., ICML 2021.
>
> [2] Self-Organized Group for Cooperative Multi-agent Reinforcement Learning, Shao et al., NIPS 2022.
>
> ---
>
> **We hope the above response can address the concerns of the reviewer. If the reviewer still has questions, please feel free to discuss with us! Thanks again to the reviewer for providing us with helpful suggestions and feedback.**

---

> ### Author Response · Authors · 2023-08-18
> **Reply to Reviewer NEb7**
>
> We further provide more ablation on alpha and also the case where the team-related information is completely removed. If our responses and experiments can address your concerns, we hope the reviewer will be willing to raise the score. If there are still remaining concerns, we are also very happy to have further discussions.

---

> > ### Comment · Reviewer_NEb7 · 2023-08-19
> > **Official Comment by Reviewer NEb7**
> >
> > Thank you for the author's response. Most of my questions have been resolved, and I have adjusted my scores accordingly.
> >
> > I kindly request that the author integrate the newly added experiments and discussions into the paper. Furthermore, I suggest adding the works I mentioned into the related works, as they have strong relevance to multi-agent systems and mutual information. This would greatly enhance the overall completeness of the paper.
> >
> > Thanks to the authors for their efforts.

---

> > > ### Author Response · Authors · 2023-08-19
> > > **Reply to Reviewer NEb7**
> > >
> > > For sure, we will add corresponding texts in the related works to reflect your suggestions. Also, added experiments and discussions will be included in the later version of our paper, as they do improve the quality of our paper.
> > >
> > > We really appreciate your time and effort. Thanks for helping us improve our paper.

---

### Author Rebuttal · Authors · 2023-08-09

We would like to thank all reviewers for their time and efforts to review our paper and give insightful comments that could further improve our work.

As several reviewers suggest including more experiments, we are now working on this. We plan to include experiments on Resource Collection (an MPE scenario used in COPA, which requires strong collaboration), and new baseline MAPPO in all scenarios, also ablation on alpha coefficient (including alpha=$\infty$ stands for the case where team-related information is completely removed). However, due to the limited computation resource, we can only release them one by one. We will keep working on them during the rebuttal and discussion to improve our work.

**1. Resource Collection**

We further evaluate LIMI on Resource Collection, which is a more challenging scenario in terms of the level of collaboration used by COPA [1]. During training, the map randomly initializes 3-5 agents, and during the evaluation, we will have 6-8 agents. For LIMI, $\alpha$ and $\alpha_i$ are fixed as 0.05. As suggested by previous research [1,2], random sub-group partitioning does not work well in Resource Collection, therefore we choose not to use it for LIMI in this experiment.

We plot the curve of training and evaluation performance in the attached pdf file. As we can see, LIMI outperforms the baselines by a large margin, which indicates that LIMI can also perform well in scenarios requiring strong collaboration.

**2. MAPPO**

As shown in the figure in the attached pdf file, MAPPO underperforms other methods in Resource Collection. We also include MAPPO as a new baseline in SMAC experiments, results are shown in the following table. In short, MAPPO does not perform very well in the setting of dynamic team composition.

| Tasks/#Agent | Algorithms |       Training         |                        |       Evaluation       |                        |
|:------------:|:----------:|:----------------------:|:----------------------:|:----------------------:|:----------------------:|
|              |            |          3-5           |           6            |           7            |           8            |
|      SZ      |    LIMI    |$0.659\pm0.02$          |$\textbf{0.453}\pm0.08$ |$\textbf{0.404}\pm0.062$|$\textbf{0.276}\pm0.076$|
|      SZ      |    REFIL   |$\textbf{0.674}\pm0.038$|$0.441\pm0.103$         |$0.352\pm0.078$         |$0.236\pm0.103$         |
|      SZ      |    AQMIX   |$0.528\pm0.044$         |$0.343\pm0.105$         |$0.291\pm0.084$         |$0.182\pm0.058$         |
|      SZ      |    CollaQ  |$0.588\pm0.03$          |$0.366\pm0.086$         |$0.314\pm0.076$         |$0.198\pm0.097$         |
|      SZ      |    MAPPO   |$0.256\pm0.01$          |$0.129\pm0.019$         |$0.148\pm0.031$         |$0.036\pm0.015$         |
|--------------|------------|------------------------|------------------------|------------------------|------------------------|
|              |            |          3-5           |           6            |           7            |           8            |
|      CSZ     |    LIMI    |$0.548\pm0.032$         |$\textbf{0.42}\pm0.102$ |$\textbf{0.297}\pm0.112$|$\textbf{0.261}\pm0.09$ |
|      CSZ     |    REFIL   |$\textbf{0.568}\pm0.027$|$0.348\pm0.057$         |$0.229\pm0.053$         |$0.164\pm0.06$          |
|      CSZ     |    AQMIX   |$0.509\pm0.054$         |$0.323\pm0.096$         |$0.216\pm0.101$         |$0.152\pm0.071$         |
|      CSZ     |    CollaQ  |$0.459\pm0.061$         |$0.362\pm0.13$          |$0.267\pm0.099$         |$0.231\pm0.095$         |
|      CSZ     |    MAPPO   |$0.248\pm0.037$         |$0.12\pm0.029$          |$0.06\pm0.028$          |$0.054\pm0.013$         |
|--------------|------------|------------------------|------------------------|------------------------|------------------------|
|              |            |          3-5           |           6            |           7            |           8            |
|      MMM     |    LIMI    |$0.548\pm0.023$         |$0.495\pm0.054$         |$\textbf{0.447}\pm0.041$|$\textbf{0.467}\pm0.067$|
|      MMM     |    REFIL   |$\textbf{0.605}\pm0.057$|$0.437\pm0.118$         |$0.329\pm0.171$         |$0.224\pm0.163$         |
|      MMM     |    AQMIX   |$0.501\pm0.036$         |$0.447\pm0.043$         |$0.344\pm0.071$         |$0.251\pm0.089$         |
|      MMM     |    CollaQ  |$0.589\pm0.027$         |$\textbf{0.513}\pm0.07$ |$0.423\pm0.026$         |$0.286\pm0.083$         |
|      MMM     |    MAPPO   |$0.289\pm0.097$         |$0.32\pm0.102$          |$0.25\pm0.063$          |$0.275\pm0.098$         |
|--------------|------------|------------------------|------------------------|------------------------|------------------------|

**3. Ablation on alpha**

To be updated during discussion once the results are available.

---

[1] Coach-player multi-agent reinforcement learning for dynamic team composition, Liu et al., ICML 2021

[2] Self-Organized Group for Cooperative Multi-agent Reinforcement Learning, Shao et al., NIPS 2022.

---

> ### Author Response · Authors · 2023-08-11
> **Ablation on alpha**
>
> We further include the ablation study on the impact of alpha. We train LIMI with different alpha on 3-5MMM and evaluate their performance on 6,7,8MMM scenarios. We use alpha=$\infty$ to represent the case where team-related information is completely removed (it is worth noting how this is different from actually set alpha=$\infty$ in LIMI). Results are summarized in the following table. As we can see, unless alpha is set unreasonably, LIMI can always achieve better generalization ability without sacrificing the training performance.
>
>
> | Tasks/#Agent |    Alpha   |       Training         |                        |       Evaluation       |                        |
> |:------------:|:----------:|:----------------------:|:----------------------:|:----------------------:|:----------------------:|
> |              |            |          3-5           |           6            |           7            |           8            |
> |      MMM     |  0 (Value) |$0.545\pm0.048$         |$0.505\pm0.058$         |$0.391\pm0.083$         |$0.319\pm0.105$         |
> |      MMM     |  0.05      |$\textbf{0.59}\pm0.008$ |$\textbf{0.59}\pm0.053$ |$\textbf{0.526}\pm0.055$|$0.426\pm0.152$         |
> |      MMM     |  0.1 (LIMI)|$0.548\pm0.023$         |$0.495\pm0.054$         |$0.447\pm0.041$         |$\textbf{0.467}\pm0.067$|
> |      MMM     |  0.5       |$0.277\pm0.042$         |$0.158\pm0.094$         |$0.18\pm0.105$          |$0.139\pm0.056$         |
> |      MMM     |  $\infty$  |$0.396\pm0.07$          |$0.432\pm0.018$         |$0.383\pm0.041$         |$0.315\pm0.061$         |
> |--------------|------------|------------------------|------------------------|------------------------|------------------------|

---

> > ### Author Response · Authors · 2023-08-18
> > **More ablation on alpha**
> >
> > We again perform the ablation study in SZ and CSZ scenarios. As we can see, given a reasonable alpha, LIMI can still perform well in terms of training performance and evaluation performance. It's worth noting that alpha=$\infty$ outperforms Value here, which further indicates that reducing the dependency on team-related information promotes generalization, even the team-related information is completely removed. However, this strategy is not widely effective and sacrifices the training performance too much in some cases (see MMM), which further leads to a decay in both training and evaluation. In contrast, our method uses alpha to control the degree of dependency on team-related information, which provides more flexibility.
> >
> > | Tasks/#Agent |    Alpha    |       Training         |                        |       Evaluation       |                        |
> > |:------------:|:-----------:|:----------------------:|:----------------------:|:----------------------:|:----------------------:|
> > |              |             |          3-5           |           6            |           7            |           8            |
> > |      SZ      |  0 (Value)  |$0.621\pm0.042$         |$0.336\pm0.075$         |$0.275\pm0.114$         |$0.154\pm0.052$         |
> > |      SZ      |  0.01       |$\textbf{0.672}\pm0.02$ |$0.394\pm0.065$         |$0.365\pm0.068$         |$0.261\pm0.092$         |
> > |      SZ      |  0.03 (LIMI)|$0.659\pm0.02$          |$\textbf{0.453}\pm0.08$ |$0.404\pm0.062$         |$0.276\pm0.076$         |
> > |      SZ      |  0.05       |$0.643\pm0.02$          |$0.447\pm0.062$         |$\textbf{0.408}\pm0.054$|$\textbf{0.313}\pm0.069$|
> > |      SZ      |  0.1        |$0.475\pm0.073$         |$0.277\pm0.125$         |$0.26\pm0.093$          |$0.146\pm0.075$         |
> > |      SZ      |  0.5        |$0.175\pm0.053$         |$0.056\pm0.015$         |$0.129\pm0.026$         |$0.043\pm0.021$         |
> > |      SZ      |  $\infty$   |$0.546\pm0.069$         |$0.429\pm0.036$         |$0.389\pm0.038$         |$0.221\pm0.004$         |
> > |--------------|------------ |------------------------|------------------------|------------------------|------------------------|
> >
> >
> > | Tasks/#Agent |    Alpha    |       Training         |                        |       Evaluation       |                        |
> > |:------------:|:-----------:|:----------------------:|:----------------------:|:----------------------:|:----------------------:|
> > |              |             |          3-5           |           6            |           7            |           8            |
> > |      CSZ     |  0 (Value)  |$0.542\pm0.059$         |$0.368\pm0.083$         |$0.207\pm0.076$         |$0.172\pm0.112$         |
> > |      CSZ     |  0.01       |$\textbf{0.592}\pm0.02$ |$0.378\pm0.033$         |$\textbf{0.364}\pm0.073$|$\textbf{0.304}\pm0.056$|
> > |      CSZ     |  0.03 (LIMI)|$0.548\pm0.032$         |$\textbf{0.42}\pm0.102$ |$0.297\pm0.112$         |$0.261\pm0.09$          |
> > |      CSZ     |  0.05       |$0.506\pm0.046$         |$0.417\pm0.092$         |$0.223\pm0.094$         |$0.192\pm0.091$         |
> > |      CSZ     |  0.1        |$0.344\pm0.05$          |$0.218\pm0.16$          |$0.113\pm0.079$         |$0.098\pm0.086$         |
> > |      CSZ     |  $\infty$   |$0.506\pm0.076$         |$0.368\pm0.064$         |$0.309\pm0.056$         |$0.27\pm0.07$           |
> > |--------------|------------ |------------------------|------------------------|------------------------|------------------------|

---

### Decision · Program_Chairs · 2023-09-21

**Decision:**

Accept (poster)

**Comment:**

The paper is about the use of mutual information as an intrinsic reward to prevent the overfitting of specific team decompositions and generalize over unseen tasks.
Although the reviewers raised some concerns about some possible side effects of the proposed approach, the lack of some related works and relevant baselines in the empirical evaluation, and some presentation issues.
Nevertheless, thanks to the authors' clarifications, the reviewers had most of their issues solved, and they found a consensus on acceptance.
The authors need to modify their paper to incorporate into the paper all the clarifications they provided during the rebuttal period and follow all the reviewers' suggestions to improve the final version of the paper.